# CD44 Receptor-Mediated/Reactive Oxygen Species-Sensitive Delivery of Nanophotosensitizers against Cervical Cancer Cells

**DOI:** 10.3390/ijms23073594

**Published:** 2022-03-25

**Authors:** Jieun Yoon, Howard Kim, Young-IL Jeong, Hoe Saeng Yang

**Affiliations:** 1Department of Medicine, Graduate School, Dongguk University, Gyeongju 38067, Korea; yje1010@yandex.com (J.Y.); howardkim83@naver.com (H.K.); 2Research Institute of Convergence of Biomedical Sciences, Pusan National University Yangsan Hospital, Gyeongnam 50612, Korea; 3The Institute of Dental Science, Chosun University, Gwangju 61452, Korea; 4Department of Obstetrics and Gynecology, Dongguk University College of Medicine, Gyeongju 38067, Korea

**Keywords:** photodynamic therapy, chlorin e6, cervical cancer, CD44 receptor, ROS-sensitive, nanophotosensitizers

## Abstract

Stimulus-sensitive, nanomedicine-based photosensitizer delivery has an opportunity to target tumor tissues since oxidative stress and the expression of molecular proteins, such as CD44 receptors, are elevated in the tumor microenvironment. The aim of this study is to investigate the CD44 receptor- and reactive oxygen species (ROS)-sensitive delivery of nanophotosensitizers of chlorin e6 (Ce6)-conjugated hyaluronic acid (HA) against HeLa human cervical cancer cells. For the synthesis of nanophotosensitizers, thioketal diamine was conjugated with the carboxyl group in HA and then the amine end group of HA-thioketal amine conjugates was conjugated again with Ce6 (Abbreviated as HAthCe6). The HAthCe6 nanophotosensitizers were of small diameter, with sizes less than 200. Their morphology was round-shaped in the observations using a transmission electron microscope (TEM). The HAthCe6 nanophotosensitizers responded to oxidative stress-induced changes in size distribution when H_2_O_2_ was added to the nanophotosensitizer aqueous solution, i.e., their monomodal distribution pattern at 0 mM H_2_O_2_ was changed to dual- and/or multi-modal distribution patterns at higher concentrations of H_2_O_2_. Furthermore, the oxidative stress induced by the H_2_O_2_ addition contributed to the disintegration of HAthCe6 nanophotosensitizers in morphology, and this phenomenon accelerated the release rate of Ce6 from nanophotosensitizers. In a cell culture study using HeLa cells, nanophotosensitizers increased Ce6 uptake ratio, ROS generation and PDT efficacy compared to free Ce6. Since HA specifically bonds with the CD44 receptor of cancer cells, the pretreatment of free HA against HeLa cells decreased the Ce6 uptake ratio, ROS generation and PDT efficacy of HAthCe6 nanophotosensitizers. These results indicated that intracellular delivery of HAthCe6 nanophotosensitizers can be controlled by the CD44 receptor-mediated pathway. Furthermore, these phenomena induced CD44 receptor-controllable ROS generation and PDT efficacy by HAthCe6 nanophotosensitizers. During in vivo tumor imaging using HeLa cells, nanophotosensitizer administration showed that the fluorescence intensity of tumor tissues was relatively higher than that of other organs. When free HA was pretreated, the fluorescence intensity of tumor tissue was relatively lower than those of other organs, indicating that HAthCe6 nanophotosensitizers have CD44 receptor sensitivity and that they can be delivered by receptor-specific manner. We suggest that HAthCe6 nanophotosensitizers are promising candidates for PDT in cervical cancer.

## 1. Introduction

Cervical cancer is derived from the cervix and most of the cases is derived from infections of human papilloma virus [1]. In accordance with pathological stage, therapeutic options may include radiotherapy, chemotherapy, surgery, and immunotherapy according to the pathological state [2,3,4,5,6,7]. However, the recurrence rate after radical hysterectomy is higher than 10% among treated patients and its 5-year survival rate is less than 5% [1,2,3]. Even though diagnosis and treatment in an early stage leads to the prevention of cervical cancers, they are frequently diagnosed in an advanced stage and are one of the key risk factors of the cancer-derived mortality of women in developing countries [8,9]. For patients in an advanced stage, chemotherapy and pelvic irradiation are frequently considered as effective treatment options, while radiotherapy or surgical removal are considered as a curative option in an early stage [9,10,11,12]. For example, chemotherapy using an once-weekly administration of cisplatin combined with radiotherapy is a typical treatment option for pelvic tumor [9,12]. Neoadjuvant chemotherapy for advanced cervical cancers has effectiveness in reducing tumor size and minimizing the risk of surgery [13]. However, median survival time after chemotherapy and/or radiotherapy still remains less than 20 months, even though these treatment regimens efficiently increase median survival time [14]. Furthermore, serious side effects against chemotherapeutic agents, such as bone marrow depression, neurotoxicity, thrombocytopenia, hematological toxicity, neutropenia, anemia and nephrotoxicity, limit the clinical application of chemotherapeutic agents [8,15,16]. Furthermore, drug resistance to chemotherapeutic agents is problematic in most cancers, including cervical cancer, and causes failure in chemotherapy [16,17]. Freitas et al., reported that a combination of PDT and a chemotherapeutic agent has a synergistic effect against cervical cancer [18]. They argued that PDT for cancer cells sensitizes cisplatin-mediated anticancer activity, while cisplatin monotherapy has limited cytotoxicity for cancer cells with severe side effects. Furthermore, PDT with 5-aminolevulinic acid (5-ALA) potentiated cisplatin cytotoxicity against HeLa human cancer cells [19].

In early stage cervical cancer, PDT leads to full human papillomavirus (HPV) elimination in more than 90% of cervical cancer patients [20]. Xu et al., also reported that PDT of cervical intraepithelial neoplasia using 5-ALA is an efficient option for diagnosis and therapy, i.e., 5-ALA treatment in a 10–30% concentration was safe for patients with negligible adverse events [21]. PDT is considered a safe option for cancer patients because it only includes photosensitizers, light and oxygen [21,22,23]. The peculiarity of PDT for cancer patients is the fact that light irradiation at a specific wavelength only activates photosensitizers to generate ROS, i.e., absence of irradiation cannot activate them, and the surrounding cells/tissues receive negligible adverse effects [21,22,23]. These advantages promote the clinical application of various photosensitizers for cancer patients [18,19,20,21,22,23,24,25,26]. PDT is a suitable treatment option for squamous cell carcinoma, such as skin cancers, cervical cancers and various epithelial cancers. PDT treatment for cervical cancers seems to be a promising candidate because more than 90% of cervical cancers are composed of squamous cell carcinoma phenotype [25,27]. However, some drawbacks of traditional photosensitizer-mediated PDT still limit clinical applications. For example, depth of light irradiation is limited less than 15 mm from the surface of tissues, and PDT is practically unable to be employed in systemic cancers [28,29]. Furthermore, the systemic administration of traditional photosensitizers such as 5-ALA leaves them practically distributed throughout the whole body, causing light sensitivity problems for patients [30]. Furthermore, the low aqueous solubility of photosensitizers and the resistance problem in cancer cells are also problematic for clinical application [31,32].

To solve obstacles of traditional photosensitizers, various delivery platforms based on nanotechnology have been investigated to improve the cancer-specific delivery of photosensitizers [33,34,35,36]. Since nano-dimensional carriers have small sizes around 100 nm, they have intrinsic characters such as huge surface area for decoration with ligands or targeting moieties, ease of solubilization of lipophilic agents, and avoidance of the reticuloendothelial system [37,38]. For example, Matlou and Abrahamse reported that hybrid inorganic–organic nanoparticles have the potential to target tumors by surface-decorated ligands and improve PDT efficacy [33]. Sun et al., reported that acid-activable peptides are ideal carriers for targeting tumors and concentrate photosensitizers in the tumor microenvironment, since the tumor microenvironment has an acidic property compared to its normal counterpart [34]. Furthermore, methoxy poly(ethylene glycol)-conjugated chlorin e6 (Ce6) can be used in the diagnosis and therapy of colon cancer cells [36].

In this study we synthesized hyaluronic acid (HA)-Ce6 conjugates via a thioketal linker (abbreviated as HAthCe6). Since cervical cancer cells such as HeLa cells express the CD44 receptor excessively, HA, which is a primary CD44-binding molecule, was used to conjugate Ce6 for the solubilization of photosensitizers and the targeting of cancer cells [39]. Furthermore, the thioketal linker was introduced between HA and Ce6 because a thioketal linker can be disintegrated by oxidative stress [40]. Nanophotosensitizers of HAthCe6 conjugates were fabricated for the PDT of cervical cancer cells. HAthCe6 nanophotosensitizers may have dual targeting properties against oxidative stress and the CD44 receptors in HeLa cells. We investigated the physicochemical and biological properties of HAthCe6 nanophotosensitizers in vitro and in vivo.

## 2. Results

### 2.1. Synthesis of HAthCe6 Conjugates

Figure 1 shows the synthesis scheme and the ^1^H NMR spectra of HAthCe6 conjugates synthesized through thioketal linkages. HA was treated with an EDAC/HOBt system to activate the carboxyl group and then conjugated with thioketal diamine to produce HA-thioketal amine conjugates, as shown in Figure 1. Following this, carboxyl group of Ce6 was also activated with an EDAC/NHS system. After that, the Ce6-NHS was conjugated again with the amine end group of HA-thioketal conjugates. As shown in Figure 1, thioketal diamine showed specific peaks between 2.0 and 4.0 ppm. Peaks of HA were confirmed at 1.6 ppm and 1.8~4.0 ppm. Ce6 peaks were shown between 1.0 and 10.0 ppm (Figure 1). In the ^1^H NMR spectra, the specific peaks of HA and thioketal linkages and Ce6 were also observed between 1.0 and 8.0 ppm, indicating that HAthCe6 conjugates were successfully synthesized. Table 1 shows the characterization of the HAthCe6 conjugates. The experimental value of the Ce6 content was 9.6% (*w*/*w*) while the theoretical value was 10.3% (*w*/*w*). These results might be because unreacted Ce6 was liberated during the dialysis procedure.

### 2.2. Nanophotosensitizer Fabrication and Acterization

HAthCe6 nanophotosensitizers were fabricated by the dialysis procedure. Figure 2 shows their morphology and particle size distribution. The average particle size of HAthCe6 nanophotosensitizers was 146.1 ± 35.3 nm and they had monomodal distribution patterns, as shown in Figure 2a. Their morphologies were spherical shapes and their diameters were smaller than 200 nm, indicating that HAthCe6 conjugates were successfully fabricated as nano-sized vehicles and have small sizes.

To assess ROS sensitivity, HAthCe6 nanophotosensitizers were incubated with hydrogen peroxide (H_2_O_2_) as shown in Figure 3. Following this, changes in the morphology and particle size distribution of the nanophotosensitizers were observed. As shown in Figure 3a–c, the monomodal distribution pattern in size distribution at 0 or 1.0 mM H_2_O_2_ was changed to a dual or multi-modal distribution pattern at higher H_2_O_2_ concentrations. At low H_2_O_2_ concentrations (1.0 mM), the size distribution of nanophotosensitizers became broader compared to the untreated samples, as shown in Figure 2a,b. H_2_O_2_ concentration higher than 5 mM resulted in dual- or multi-modal distribution patterns, i.e., size distribution was dual-distribution pattern at 5.0 mM H_2_O_2_ and, at 10 mM H_2_O_2_, measurement was practically failed (Figure 3b,c). These results might be due to the thioketal linkage between HA and Ce6 which must have been broken at low H_2_O_2_ concentration and then the Ce6 was separated from the HAthCe6 conjugates. These must be resulted in an increase of particle size distribution. Furthermore, they must be disintegrated at higher H_2_O_2_ concentration by the liberation of Ce6 from the nanophotosensitizers. Morphological observation supported these results, as shown in Figure 3d–f. The morphologies of nanophotosensitizers were changed compared to untreated samples, as shown in Figure 2b, i.e., some of the nanophotosensitizers swelled and/or disintegrated at low H_2_O_2_ concentrations (1.0 mM), even though most of them still maintained their spherical morphology (Figure 3d). Furthermore, nanophotosensitizers were largely swelled or disintegrated at 5 mM H_2_O_2_, and at 10 mM H_2_O_2_, most of them disintegrated, as shown in Figure 3f, indicating that HAthCe6 nanophotosensitizers have ROS-sensitive disintegration properties. Table 2 provides abbreviated details of the particle size distribution properties presented in Figure 3a–c.

Figure 4 shows H_2_O_2_ concentration’s effect on the changes to the Ce6 release rate and fluorescence intensity. The Ce6 release rate was very low at 0 mM H_2_O_2_ concentration (Figure 4a). However, increase of H_2_O_2_ concentration in the release media significantly increased Ce6 release rate from the nanophotosensitizers. Furthermore, fluorescence intensity of nanophotosensitizer solution also increased according to the increase of H_2_O_2_ concentration (Figure 4b), indicating that HAthCe6 nanophotosensitizers responded to ROS and that Ce6 release can be controlled by H_2_O_2_ concentration.

### 2.3. Cell Culture Study and PDT In Vitro

Figure 5 shows Ce6 uptake ratio, as evaluated with HeLa human cervical cancer cells. According to the increase in Ce6 concentration, intracellular Ce6 uptake ratio by HeLa cells was gradually elevated both with Ce6 itself and HAthCe6 nanophotosensitizers as shown in Figure 5a. As shown in Figure 5b, nanophotosensitizer treatment showed higher red fluorescence intensity than Ce6 treatment. These results indicated that nanophotosensitizers have superior potential in targeting cancer cells over Ce6 alone. To assess whether or not HAthCe6 nanophotosensitizers were able to be delivered through the CD44 receptor-mediated pathway, HA was pretreated in HeLa cells (Figure 5 and Appendix A). When HA was pretreated in HeLa cells, the Ce6 uptake ratio was significantly inhibited. Furthermore, a higher HA concentration induced a lower Ce6 uptake ratio of HAthCe6 nanophotosensitizers, indicating that HAthCe6 nanophotosensitizers can be delivered by a CD44-mediated pathway, i.e., intracellular delivery can be controlled by blocking the CD44 receptor.

Figure 6 shows the dark toxicity of HAthCe6 nanophotosensitizers against normal cells (RAW264.7 cells, Figure 6a) and cancer cells (HeLa cells, Figure 6b). Ce6 showed only small toxicity in the dark condition until 5 µg/mL, i.e., RAW264.7 cells showed higher cell viability than 80% of both free Ce6 and the nanophotosensitizers. As well as free Ce6, nanophotosensitizers did not significantly affect to the viability of HeLa cells and 80% of cells were viable until 5 µg/mL (Ce6 equivalent). Interestingly, HeLa cell viability was decreased to 73% of free Ce6 treatment at 5 µg/mL, while the nanophotosensitizer treatment was maintained at above 80% at 5 µg/mL. These results indicated that nanophotosensitizers have reduced cytotoxicity to both RAW264.7 and HeLa cells, similarly to free Ce6. Furthermore, nanophotosensitizers have no acute cytotoxicity against normal cells or cancer cells.

Figure 7 shows the ROS generation and PDT efficacy in HeLa cells. Figure 7a shows ROS generation and Figure 7b showed PDT efficacy in HeLa cells by treatment of HAthCe6 nanophotosensitizers. ROS generation in HeLa cells was gradually elevated according to Ce6 concentration in both free Ce6 and HAthCe6 nanophotosensitizers. ROS generation was significantly higher in nanophotosensitizer treatment compared to free Ce6 treatment. These results indicated that nanophotosensitizers had higher intracellular delivery and ROS formation in cancer cells. As expected, PDT efficacy was also significantly increased by treatment with nanophotosensitizers, i.e., the viability of HeLa cells by treatment with free Ce6 was higher than 80% at 1 µg/mL Ce6 concentration, while nanophotosensitizers resulted in less than 30% cell viability (Figure 7b), indicating that ROS generation and PDT efficacy of nanophotosensitizers in HeLa cells was superior than that of Ce6 alone. Furthermore, the effect of HA pretreatment to block CD44 receptors in HeLa cells was also evaluated through CD44 receptor-mediated ROS generation and the PDT efficacy of nanophotosensitizers (Figure 7). Figure 7a shows that ROS generation was significantly decreased by pretreatment with HA. Furthermore, higher HA concentrations induced lower ROS generation, i.e., the ROS level at 2 µg/mL was decreased less than 600 (a.u.) by pretreatment with HA (5.0 mg/mL) while ROS levels were higher than 1200 without pretreatment with HA (0 mg/mL). These results indicated that the ROS generation capacity of nanophotosensitizers can be controlled by their CD44 receptor-mediated delivery capacity. As expected, PDT efficacy was also decreased by pretreatment with HA, i.e., cell viability at a 1.0 µg/mL Ce6 concentration was less than 30% in the absence of HA pretreatment (0 mg/mL) while HA pretreatment (5.0 mg/mL) resulted in an increase of cell viability above 70%, indicating that nanophotosensitizers have superior CD44 receptor responsiveness in ROS generation and PDT efficacy. These means that the oxidative stress and PDT efficacy of nanophotosensitizers can be controlled by the CD44 receptors of cancer cells.

### 2.4. In Vivo Animal Tumor Imaging

Figure 8 shows the CD44 receptor-mediated delivery capacity of nanophotosensitizers using a tumor xenograft model. The tumor xenograft model was prepared by subcutaneous injection of HeLa cells to the back of mice. Prior to injection of nanophotosensitizers, free HA was intravenously (i.v.) administered to block the CD44 receptor of the tumor. Fluorescence was elevated in tumor tissues rather than other organs in the absence of free HA pretreatment (Free HA, 0 mg/kg), indicating that HAthCe6 nanophotosensitizers have superior potential in HeLa tumor targeting. Furthermore, fluorescence intensity in tumor tissue was decreased lower than other organs by pretreatment of free HA (10 mg/kg), indicating that the delivery of HAthCe6 nanophotosensitizers can be controlled by CD44 receptors.

## 3. Discussion

Compared to normal tissues, microenvironments of tumor tissues are quite different [41,42,43]. Due to the complexity of tumor microenvironment, a therapeutic strategy should be established based on the physiological/biological status of the tumor microenvironment [42]. One of the key features of a tumor microenvironment is an elevated level of redox-related factors [43]. Wu et al., reported that an elevated redox score is correlated with an increase in the tumor mutation burden and driver gene mutation rates [43]. They argued that higher redox potential has a relationship with the sensitivity/resistance to anticancer drugs and poor prognosis. Furthermore, the aggressiveness of a tumor contributes to the imbalance to the redox state of cancer cells and some of chemical agents that contribute to the generation of ROS, which may aggravate aggressiveness and complicate tumor therapy [44]. Paradoxically, an imbalance of the redox status of a tumor microenvironment can be used for targeted therapy of cancer. Glass et al., reported that a nanoparticle adelivery system can be altered to be sensitive to the redox status of a tumor to produce ROS by photoactivation and/or radiation therapy via enhancing ROS production [45]. For example, ROS-producing agents such as piperlongumine elevate oxidative stress in cancer cells and then induce ROS-mediated cancer cell death via a synergistic reaction with traditional anticancer drugs [46]. Photosensitizers for PDT are typical ROS-producing agents upon light irradiation [18,19]. ROS can be produced by photosensitizers excessively in the field of light irradiation and then induce death in cancer cells through oxidative stress [47]. Due to these advantages, PDT has been extensively investigated for several decades due to its safety in cancer patients [18,19,20,21,22,23,24,25,26,28,29,30,31,32,33,34,35,36]. However, deficiency in the cancer-specificity of traditional photosensitizers may arouse photosensitivity [30]. Nanoparticles can be used as a solution to overcome these obstacles, since nanoparticles enable anticancer agents to direct tumor tissues through their long-circulation and/or tumor-specific delivery capacities [48,49]. ROS-mediated degradable nanoparticles may synergize with PDT in cancer because PDT induces ROS levels in cancer cells and accelerates the degradation of nanoparticles [50]. Sun et al., reported that pyropheophorbide a-based PDT using ROS-sensitive nanoassemblies synergized with the paclitaxel-based chemotherapy of cancer cells through the ROS-specific release of paclitaxel [50]. In our results, H_2_O_2_ induced disintegration of HAthCe6 nanophotosensitizers and the Ce6 release rate was accelerated, as shown in Figure 3 and Figure 4. Oxidative stress may degrade the thioketal linker of HAthCe6 nanophotosensitizers and then liberate Ce6. Sun et al., also reported that a hyperbranched polyphosphate containing a thioketal linker can be disintegrated by light irradiation and accelerate doxorubicin release rate [51]. Our results also showed that a thioketal linker between HA and Ce6 might be disintegrated in a low concentration of H_2_O_2_ and the particle size distribution was increased, as shown in Figure 3a. At higher H_2_O_2_ concentrations, Ce6 might be separated from the HA backbone of HAthCe6 conjugates and then liberated from nanophotosensitizers, as shown in Figure 3 and Figure 4. Chen et al., also reported that oxidative stress resulted in changes to the size distribution of thioketal nanoparticles, i.e., the size distribution of nanoparticles was changed from a monomodal distribution pattern to a multimodal distribution pattern by the addition of H_2_O_2_ [52]. Furthermore, they also showed that H_2_O_2_ treatment induces the degradation of nanoparticles and the drug release rate was significantly increased. The fact that the ROS level is normally elevated in tumor tissue can be used as a targeting issue, and this status induces the tumor-specific degradation of nanoparticles to liberate anticancer agents [53,54].

Meanwhile, one of the intrinsic properties of cancer cells, as distinguished from normal cells, is their abundant expression of various molecular receptors, such as CD44 receptor and folate receptor [55,56]. Son et al., reported that HA-decorated nanoparticles deliver anticancer agents in a CD44 receptor-mediated manner and induce preferential death in CD44 receptor-positive cancer cells [57]. They argued that HA-decorated nanoparticles differently inhibited the viability of cancer cells based on CD44 receptors, i.e., blocking of the CD44 receptor of HepG2 cells (a CD44 positive cell) resulted in a decrease of anticancer activity while nanoparticles did not significantly affect the death of CT26 cells (a CD44-negative cell). Furthermore, HA-coated nanoparticles having disulfide linkages were delivered through CD44 receptors of cancer cells and then nanoparticles were degraded under redox states [58]. Our results also showed that intracellular delivery of HAthCe6 nanophotosensitizers was inhibited by blocking of the CD44 receptor of cancer cells, as shown in Figure 5. These phenomena induced CD44 receptor-dependent ROS generation and the phototoxicity of HAthCe6 nanophotosensitizers in HeLa cells, as shown in Figure 7. These results indicated that HAthCe6 nanophotosensitizers has responsiveness against the CD44 receptors of HeLa cells and that therapeutic potential can be controlled by receptor expression. Furthermore, the CD44 receptor-mediated delivery of HAthCe6 nanophotosensitizers was also inhibited by the CD44-receptor blocking of HeLa tumors in an in vivo animal model, as shown in Figure 8. Our results also showed that HAthCe6 nanophotosensitizers can be delivered in ROS-sensitive and CD44 receptor-mediated manners. Nanophotosensitizers were efficiently concentrated in the tumor tissue, i.e., the fluorescence intensity from the i.v. administration of nanophotosensitizers was strongest in tumor tissue, as shown in Figure 8. This peculiarity of HAthCe6 nanophotosensitizers may alleviate the light sensitivity of normal tissues and PDT efficacy against tumors. HA-decorated nanoparticles can be delivered to cancer cells in a CD44 receptor-specific manner and delivering capacity was accelerated against CD44-receptor over-expressing cells [59]. Kim et al., also HA-decorated nanophotosensitizers selectively targeted to CD44-receptor positive cells and killed the cancer cells in a CD-responsive manner [60]. They argued that the PDT efficacy of HA-decorated nanophotosensitizers was controlled by CD44 receptor expression, while CD44 receptor-negative cells were not affected by the blocking of CD44 receptors. The delivery capacity and PDT efficacy of our HAthCe6 nanophotosensitizers were also easily controlled by CD44-receptor positive cells. These results indicated that HAthCe6 nanophotosensitizers can be used for specific PDT of cervical cancer with minimization of side-effects against normal cells, since cancer cells have overexpressed CD44 receptors [61].

## 4. Materials and Methods

### 4.1. Chemicals

HA of 5000 g/mol molecular weight was purchased from Lifecore Biomedical (Chaska, MN, USA). Ce6 was obtained from Frontier Sci. Co. (Logan, UT, USA). 3-(4,5-dimethyl-2-thiazolyl)-2, 5-diphenyl-2H-tetrazolium bromide (MTT), 2′,7′-dichlorofluorescin diacetate (DCFH-DA), hydrogen peroxide (H_2_O_2_), N-hydroxy succinimide (NHS), N-(3-dimethylaminopropyl)-N’-ethylcarbodiimide HCl (EDAC), 1-hydroxybenzotriazole (HOBt) and Cremophor^®^ EL were purchased from Sigma-Aldrich Chem. Co. (St. Louis, MI, USA). Thioketal diamine was purchased from RuixiBiotech Co. Ltd. (Xi’an, China). Dialysis membranes (Spectra/Por^®^7 Membranes, molecular weight cutoff size (MWCO): 1000 and 2000 g/mol) were purchased from Spectrum Lab., Inc. (Rancho Dominguez, CA, USA).

### 4.2. Synthesis of HAthCe6 Conjugates

The HA-thioketal amine conjugates: 400 mg of HA (≈1.0 mM as a disaccharide unit) in 10 mL DMSO/H_2_O (8/2, *v*/*v*) were mixed with 19.2 mg EDAC (0.1 mM) and 13.5 mg of HOBt (0.1 mM). This solution was reacted for 6 h and then 195 mg of thioketal diamine (1 equivalent mole of disaccharide unit in HA; 10 equivalents mole of EDAC/HOBt activated unit) was added, followed by magnetic stirring for 24 h. The resulting solution was dialyzed against water for 2 days using a dialysis tube (MWCO = 1000 g/mol). Water was exchanged every 3 h interval. The dialyzed solution was then lyophilized over 2 days to obtain HA-thioketal amine conjugates.

For the synthesis of HAthCe6 conjugates, Ce6 (12 mg) in 10 mL DMSO was mixed with 3.84 mg EDAC and 2.3 mg of NHS to activate the carboxylic acid of Ce6. Then, HA-thioketal amine (104 mg) in 10 mL DMSO/H_2_O (8/2, *v*/*v*) was introduced into the solution of Ce6 reaction, followed by magnetic stirring for 24 h in the dark condition. This solution was dialyzed using a dialysis tube (MWCO = 2000 g/mol) against water for 2 days. The water was exchanged every 3 h interval to remove organic solvents. Following this, the dialyzed solution was lyophilized over 2 days. The yield of HAthCe6 conjugates measured by weight and was higher than 94%. Yield = (weight of HAthCe6 conjugates/(weight of HA-thioketal amine + weight of Ce6)] 100.

### 4.3. ^1^H Nuclear Magnetic Resonance (NMR) Spectra

^1^H NMR spectra (Varian Unity Inova 500 MHz NB High Resolution Fourier transform (FT)-nuclear magnetic resonance (NMR) spectrometer, Varian Inc., Santa Clara, CA, USA) were employed to monitor the synthesis of the HAthCe6 conjugates. D_2_O, DMSO or D_2_O/DMSO mixtures was used to dissolve chemicals.

### 4.4. Fabrication of Nanophotosensitizers of HAthCe6 Conjugates

HAthCe6 conjugates (20 mg) in 5 mL DMSO/water mixtures (3/2, *v*/*v*) were dropped into 10 mL water to form nanoparticles. This solution was dialyzed using dialysis tube (MWCO: 2000 g/mol) against 1L deionized water for 1 day. Water was exchanged every 3 h intervals. This was used to analyze or to assess PDT efficacy. To evaluate Ce6 contents, nanophotosensitizers (5 mg/5 mL water) were mixed with 45 mL phosphate buffered saline (0.01 M, pH 7.4) (PBS). H_2_O_2_ was added to this solution (final concentration of H_2_O_2_: 100 mM) and then it was stirred magnetically for 48 h. This solution was diluted with DMSO 10 times. A fluorescence spectrophotometer (λ_ex_ = 407; λ_em_ = 664 nm) (RF-5301PC, Kyoto, Japan) or UV-VIS spectrophotometer (664 nm) (Thermo Fisher Scientific, Waltham, MA, USA) was used to measure Ce6 concentration.

To determine the calibration curve of Ce6, Ce6 (10 mg) dissolved in 10 mL DMSO (1 mg/mL) was diluted 100 times with a DMSO/water mixed solvent (9/1, *v*/*v*) with H_2_O_2_ (final concentration: 10 mM). For the calibration curve, the absorbance of this solution was measured with a Genesys 10s UV-VIS spectrophotometer. The calibration curve was measured at the range of 0.1~7 µg/mL of Ce6 concentration (Appendix A).

Ce6 content (%, *w*/*w*) = (measured weight of Ce6/total weight of nanophotosensitizers)/100. The Ce6 content in the nanophotosensitizers made from the HAthCe6 conjugates was approximately 9.6% (*w*/*w*).

### 4.5. Transmission Electron Microscope (TEM)

TEM (H-7600, Hitachi Instruments Ltd., Tokyo, Japan) was employed to observe the morphology of HAthCe6 nanophotosensitizers. Nanophotosensitizer solution was placed onto the carbon film-coated grid and then dried at room temperature. For negative staining, 10 µL of phosphotungstic acid was added to 100 µL of nanophotosensitizer solution. Nanophotosensitizer observation was carried out at 80 kV.

### 4.6. Fluorescence Spectrophotometer Measurement

The fluorescence property of aqueous solution of HAthCe6 nanophotosensitizers was measured using a fluorescence spectrofluorophotometer (RF-5301PCspectrofluorophometer, Kyoto, Japan). To react with H_2_O_2_, the Ce6 concentration was adjusted to 0.1 mg/mL in PBS and then incubated at 37 °C for 4 h. This solution was scanned between 500 nm and 800 nm (λ_ex_ = 400 nm). Fluorescence images were also observed with a Maestro^TM^ 2 small animal imaging instrument, Cambridge Research and Instrumentation Inc. (Hopkinton, MA 01801, USA).

### 4.7. Drug Release from Nanophotosensitizer

Nanophotosensitizers were reconstituted in PBS for the drug release study. H_2_O_2_ was added to the media to study the effect of ROS on the drug release rate. A total of 5 mL of nanophotosensitizers (1 mg nanophotosensitizers/mL PBS) was put into the dialysis tube (MWCO = 2000 g/mol). Then, the dialysis tube was introduced into a 50 mL conical tube with 45 mL PBS. Following this, they were incubated with an SI-600R shaker incubator (Jeiotech Co., Daejeon, Korea) at 37 °C and 100 rpm. The media were replaced with fresh media at predetermined time intervals. Liberated Ce6 in the media was measured with a fluorescence spectrofluorophotometer (RF-5301PC spectrofluorophometer, Kyoto, Japan) (λ_ex_ = 407; λ_em_ = 664 nm) or a UV-VIS spectrophotometer (664 nm) (Thermo Fisher Scientific, Waltham, MA, USA). All of the results were expressed as mean ± standard deviation (S.D.) from three separated experiments.

### 4.8. Cell Culture

RAW264.7 mouse macrophage and HeLa human cervical cancer cells were purchased from Korean Cell Line Bank (Seoul, Korea). MEM medium (Gibco, Grand Island, NY, USA) and Dulbecco’s modified eagle medium (DMEM) (Gibco, Grand Island, NY, USA) were used to culture the HeLa cells and RAW264.7 cells, respectively. A total of 10% heat-inactivated fetal bovine serum (FBS) (Invitrogen) and 1% penicillin/streptomycin were added to the culture media.

### 4.9. PDT of Cancer Cells

Phototoxicity (PDT): 2 × 10^4^ HeLa cells/well in 96-well plates were exposed to free Ce6 or HAthCe6 nanophotosensitizers. Ce6 in DMSO was diluted with serum-free media for treatment of free Ce6. HAthCe6 nanophotosensitizers reconstituted in water was filtered with a syringe filter (0.8 μm, Acrodisc^®^ Syringe filters, Pall CO., Cornwall, UK) for sterilization. Cells were treated with free Ce6 or nanophotosensitizers for 2 h and then washed with PBS. Following this, 100 µL phenol red-free media were added and then cells were irradiated at 664 nm (2.0 J/cm^2^) with an expanded homogenous beam from SH systems (Gwangju, Korea). A photo-radiometer (DeltaOhm, Padova, Italy) was used to measure the dose of light at 664 nm. After that, HeLa cells were further incubated in 5% CO_2_ at 37 °C for 24 h. An MTT proliferation assay was employed to measure cell viability. A total of 30 µL of MTT solution (5 mg/mL in PBS) was added, incubated for 3 h, replaced with DMSO (100 µL) and then we measured its absorbance with an Infinite M200 pro microplate reader at 570 nm. All experiments were performed in the dark condition.

Dark toxicity: cells were treated with free Ce6 or the HAthCe6 nanophotosensitizers described above without light irradiation for dark toxicity. Then, cell viability was measured with an MTT proliferation assay.

### 4.10. Intracellular Uptake of Free Ce6 or Nanophotosensitizers

A total of 2 × 10^4^ HeLa cells seeded in 96-well plates were cultured overnight at 37 °C in 5% CO_2_. Cells were treated with Ce6 or HAthCe6 nanophotosensitizers for 2 h and then the cells were washed with PBS. These were lysed to measure intracellular Ce6 levels. GenDEPOT lysis buffer (100 µL) (Barker, TX, USA) was used for lysis of cells. An Infinite M200 pro microplate reader, Tecan Trading AG (Männedorf, Switzerland) was used to measure Ce6 (λ_ex_ = 407; λ_em_ = 664 nm).

To study the effect of CD44 receptor blocking, free HA (1.0 or 5.0 mg/mL) was pretreated to HeLa cells 30 min before nanophotosensitizer treatment. After that, the cells were treated with Ce6 or nanophotosensitizers for 2 h and then lysed, as described above. Fluorescence intensity was measured at the 407 nm excitation wavelength and 664 nm emission wavelength using an Infinite M200 pro microplate reader.

### 4.11. Fluorescence Microscopy

HeLa cells were treated with free Ce6 or HAthCe6 nanophotosensitizers (3 × 10^5^ cells/well in 6 well plates with cover glass) for 90 min. Following this, supernatants were discarded and then the cells washed with PBS. Cells were fixed with paraformaldehyde solution (4%) for 15 min and then washed with PBS again. Following this, cells were immobilized with Immunomount mounting solution, Thermo Electron Co. (Pittsburgh, PA, USA). Cells were observed with an Eclipse 80i fluorescence microscope (Nikon, Tokyo, Japan).

### 4.12. ROS Generation by Ce6 or Nanophotosensitizers

Similar to PDT treatment, cells were treated with free Ce6 or HAthCe6 nanophotosensitizers (2 × 10^4^ HeLa cells/well in 96-well plates). For measurement of intracellular ROS levels, DCFH-DA reagent (final concentration: 20 µM) was also added to the cells. Cells were incubated for 2 h at 37 °C in 5% CO_2_ and then washed with PBS. After that, 100 µL fresh phenol red free media was added and irradiated at 664 nm (light dose: 2.0 J/cm^2^). Intracellular ROS levels were measured with an Infinite M200 pro microplate reader (λ_ex_ = 485; λ_em_ = 535 nm).

### 4.13. In Vivo Fluorescence Imaging and PDT Study

For in vivo fluorescence imaging of HeLa tumor mice, 1 × 10^6^ HeLa cells were subcutaneously implanted in the back of mice (nude BAL b/C mice, 20 g, 5 weeks old). Nanophotosensitizer solution was filtered with syringe filters (0.8 µm) and then intravenously (i.v.) administered through the tail vein (injection volume: 100 µL). A total of 24 h later, the mice were sacrificed to take organs including tumor tissue. A Maestro^TM^ 2 small animal imaging instrument was used to observe the biodistribution of nanophotosensitizers in mice.

The protocol of the animal experiment was followed to the guidelines of the Pusan National University Institutional Animal Care and Use Committee (PNUIACUC). The ethical procedures and scientific care protocols of animal study were reviewed and monitored by the PNUIACUC (Approval Number: PNU-2020-2751).

### 4.14. Statistical Analysis

The results are expressed as mean ± standard deviation (SD) of three experiments. The statistical analysis of the results was evaluated with a Student’s *t* test using SigmaPlot^®^ software, version: 11.0, Systat Software, Inc. Co. (San Jose, CA, USA) or a one-way analysis of variance (ANOVA) followed by the Tukey test using GraphPad Prism 9 (GraphPad Software LLC., San Diego, CA, USA). Then, *p* < 0.05 was evaluated as the minimal level of significance.

## 5. Conclusions

For ROS and CD44 receptor-sensitive delivery of photosensitizers, HAthCe6 nanophotosensitizers were synthesized. Thioketal diamine was attached to the carboxyl group of HA and then the amine end groups of HA-thioketal amine conjugates were conjugated with Ce6. The morphology and size of HAthCe6 nanophotosensitizers were a spherical shape and a small diameter of less than 200 nm. Oxidative stress by H_2_O_2_ addition to an aqueous nanophotosensitizer solution resulted in disintegration of the nanophotosensitizers and the monomodal distribution pattern of nanophotosensitizers were changed to dual modal or multimodal distribution. The addition of H_2_O_2_ also accelerated Ce6 release from the nanophotosensitizers, indicating that oxidative stress induces disintegration of the HAthCe6 nanophotosensitizers and then controls the Ce6 release rate. When nanophotosensitizers were treated, intracellular Ce6 uptake and ROS generation of HeLa cells were increased compared to that of free Ce6. These phenomena also induced the PDT efficacy of nanophotosensitizers. The CD44 receptors of the target cells were blocked by pretreatment of HA against HeLa cells. These induced a decrease in ROS formation and PDT efficacy, indicating that CD44 receptors affected the delivery of HAthCe6 nanophotosensitizers to HeLa cells and also affected the intracellular ROS generation or PDT efficacy. An animal imaging study using HeLa tumor-bearing mice showed that higher fluorescence intensity in tumor tissues compard to other organs was observed by the administration of nanophotosensitizers. Furthermore, the pretreatment with free HA that induced fluorescence intensity in tumor tissue was decreased compared to other organs, indicating that CD44 receptors govern the delivery capacity of HAthCe6 nanophotosensitizers in vitro and in vivo. We suggest that HAthCe6 nanophotosensitizers are promising candidate for PDT of HeLa cells.

## Figures and Tables

**Figure 1 ijms-23-03594-f001:**
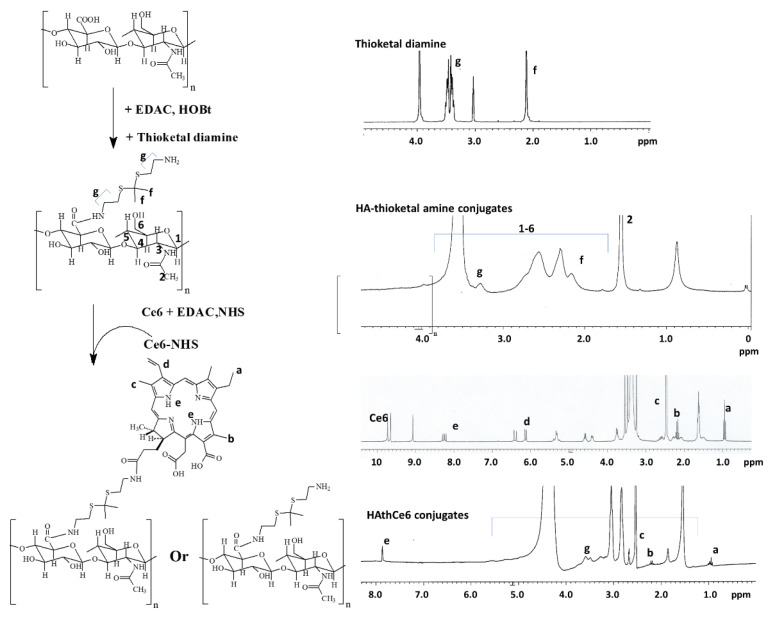
Synthesis scheme (**left**) and ^1^H NMR spectra (**right**) of HAthCe6 conjugates. The dimethyl sulfoxide (DMSO)-d form was used to dissolve Ce6 or thioketal diamine, and dimethyl sulfoxide (DMSO)-d form/D_2_O (4/1, *v*/*v*) mixtures were used to dissolve HA or HAthCe6 conjugates.

**Figure 2 ijms-23-03594-f002:**
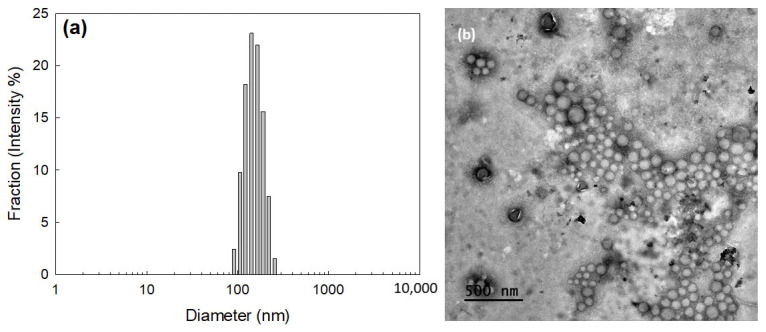
HAthCe6 nanophotosensitizers. (**a**) Particle size distribution and (**b**) TEM image. Average particle size was 146.1 ± 35.3 nm, as described in Table 1. Poly-dispersity index (PDI) was 0.036. For TEM images, nanophotosensitizers were negatively stained with phosphotungstic acid.

**Figure 3 ijms-23-03594-f003:**
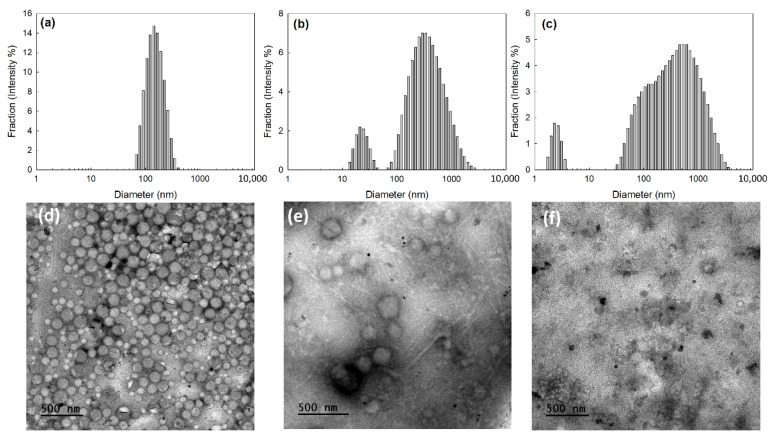
The effect of H_2_O_2_ concentration on the changes in particle size distribution (**a**–**c**), and TEM images (**d**–**f**) of HAthCe6 nanophotosensitizers: (**a**,**d**), 1.0 mM H_2_O_2_; (**b**,**e**), 5.0 mM H_2_O_2_; (**c**,**f**), 10 mM H_2_O_2_.

**Figure 4 ijms-23-03594-f004:**
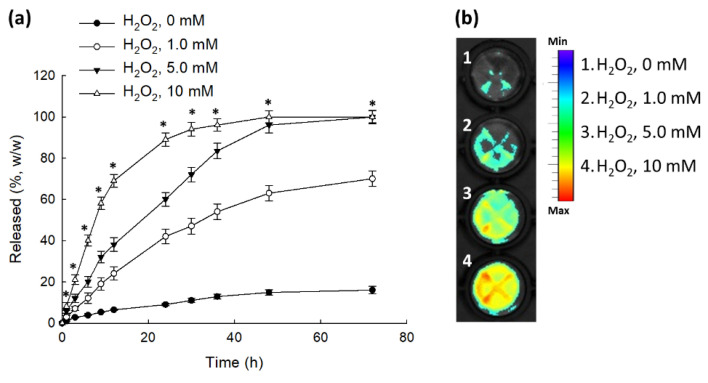
The effect of H_2_O_2_ addition on the aqueous nanophotosensitizer solution. (**a**) Ce6 release from nanophotosensitizers. (**b**) Changes to fluorescence spectra of nanophotosensitizers in aqueous solution. For measurement of fluorescence intensity, the concentration of the nanophotosensitizer solution was adjusted to 1.0 mg/mL in PBS. H_2_O_2_ was added to release media for the Ce6 release study. For fluorescence imaging, the nanophotosensitizer solution was incubated for 2 h at 37 °C in the absence or presence of different concentration of H_2_O_2_. Ce6 concentration in the release media was calculated from the calibration curve of Ce6 (Appendix A). * indicates statistical significance when H_2_O_2_ (10 mM) was compared to the H_2_O_2_ (0 mM) (ANOVA followed by Tukey’s test, *p* < 0.05).

**Figure 5 ijms-23-03594-f005:**
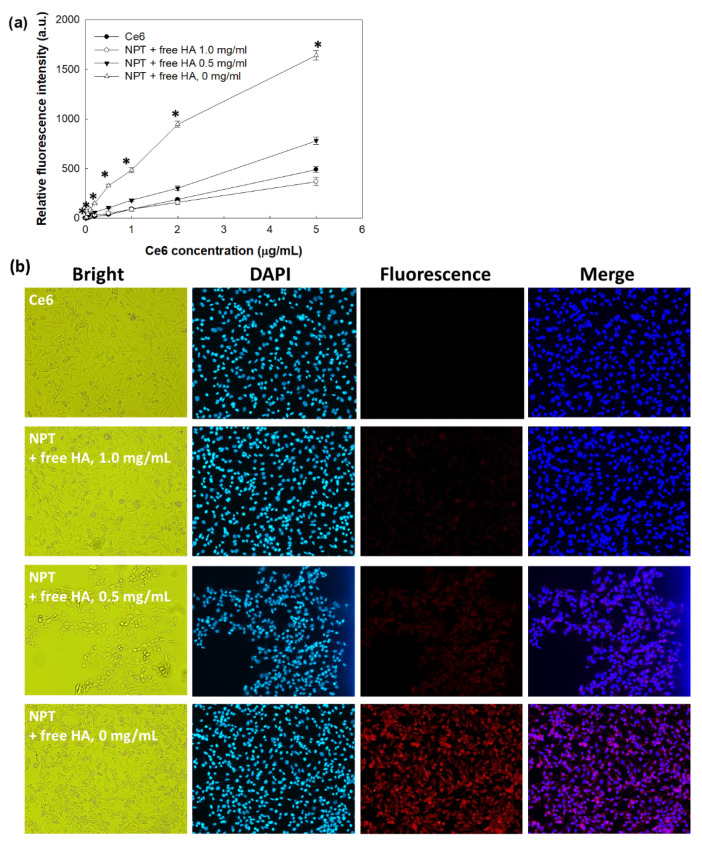
Ce6 uptake ratio of HeLa cells. (**a**) HA pretreatment effect on the Ce6 uptake ratio. (**b**) Fluorescence images of HeLa cells. Free HA (1.0 or 5.0 mg/mL) was pretreated in HeLa cells 30 min before treatment with nanophotosensitizers. Cells were treated with Ce6 or nanophotosensitizers for 2 h, and then lysed to measure intracellular Ce6 level. Fluorescence intensity was measured at the 407 nm excitation wavelength and 664 nm emission wavelength using an Infinite M200 pro microplate reader. Magnification of images: 100×. * indicates statistical significance when (NPT + free HA, 0 mg/mL) was compared to Ce6 (ANOVA followed by Tukey’s test, *p* < 0.05).

**Figure 6 ijms-23-03594-f006:**
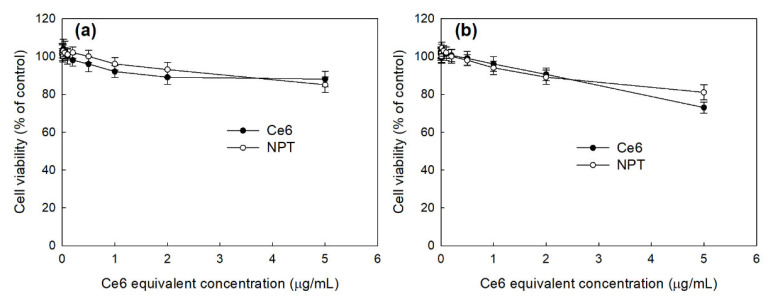
Dark toxicity of free Ce6 and nanophotosensitizers. (**a**) RAW264.7 cells; (**b**) HeLa cells. Cells were treated with free Ce6 or HAthCe6 nanophotosensitizers for 2 h without light irradiation. For control treatment, serum-free media were used for comparison.

**Figure 7 ijms-23-03594-f007:**
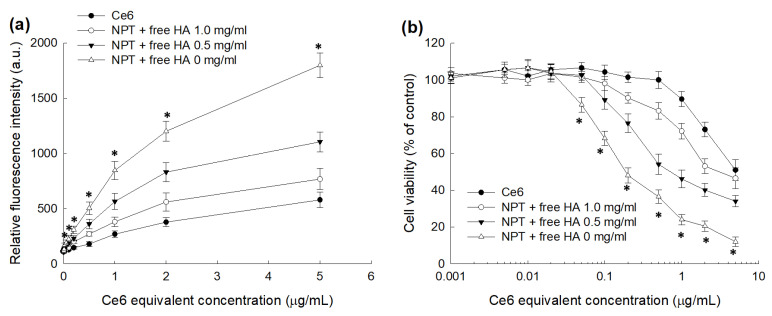
The effect of HAthCe6 nanophotosensitizers against HeLa cells. Free Ce6 or nanophotosensitizers in serum-free media were treated to 2 × 10^4^ cells/well in 96 well plates and then irradiated at a light dose of 2 J/cm^2^. (**a**) ROS generation. A DCFH-DA assay was used to evaluate intracellular ROS level. For control treatment, serum-free media were used. (**b**) PDT efficacy. Cell viability was evaluated by MTT assay. * indicates statistical significance when the group of (NPT + free HA, 0 mg/mL) was compared to the group of Ce6 (ANOVA followed by Tukey’s test, *p* < 0.05).

**Figure 8 ijms-23-03594-f008:**
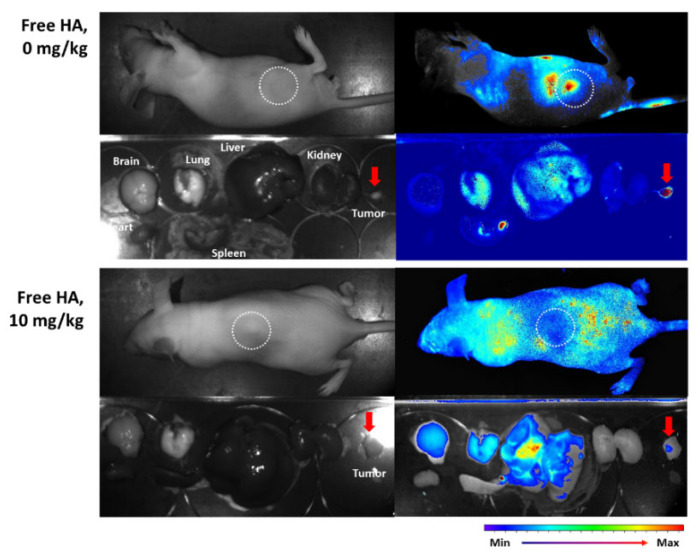
In vivo tumor imaging of HeLa tumor-bearing mice. Thirty minutes before administration of nanophotosensitizers, free HA (10 mg/kg) was i.v. administered, via tail vein, to mice. Then, nanophotosensitizers (10 mg/kg) were i.v. administered. A total of 24 h later, mice were sacrificed to observe fluorescence images of their organs using a fluorescence imaging device (Maestro II^®^, Cambridge Research and Instrumentation Inc., Hopkinton, MA, USA).

**Table 1 ijms-23-03594-t001:** Characterization of HAthCe6 conjugates.

	Drug Contents (%, *w*/*w*)		Particle Size (nm)
	Theoretical ^a^	Experimental ^b^	
HAthCe6 conjugates	10.3	9.3	146.1 ± 35.3

^a^ Theoretical content was calculated from the feeding weight of Ce6. ^b^ Experimental content was measured as depicted in the Section 4 and the Appendix A. Ce6 concentration in the release media was calculated from the calibration curve of Ce6 (Appendix A).

**Table 2 ijms-23-03594-t002:** Effect of H_2_O_2_ concentration.

H_2_O_2_ Concentration (mM)	Particle Size Distribution (nm) ^a^	Fraction(Intensity, %) ^a^	PDI
1	157.8 ± 58.95	100	0.114
5	454.6 ± 349.3	88.9	0.643
	23.61 ± 6.471	11.1	
10	550.6 ± 554.0	93.4	0.636
	2.512 ± 0.4898	6.6	

^a^ Particle size distribution and its fraction were the results of Figure 3a.

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
