# Peer review of "CD44 Receptor-Mediated/Reactive Oxygen Species-Sensitive Delivery of Nanophotosensitizers against Cervical Cancer Cells"

_ijms, 2022, doi:10.3390/ijms23073594_

Round 1

Reviewer 1 Report

In the manuscript IJMS-1591422 by Yoon et al, authors have synthesized HAthCe6, a chlorin e6 (Ce6)-conjugated with hyaluronic acid (HA) and demonstrated its efficacy for reactive oxygen species (ROS)-sensitive and CD44 receptor-targeted delivery of photosensitizers and for photodynamic therapy (PDT) of HeLa cervical carcinoma cells. In this study, authors investigated physicochemical and biological properties of HAthCe6 nano-photosensitizers both in in vitro and in vivo settings. Authors found that the uptake ratio and efficacy of photodynamic therapy of conjugated Ce6 was significantly higher than the free Ce6. Authors found HAthCe6 nano-photosensitizers effective against HeLa cells and proposed the use of conjugated (HAthCe6) nano-photosensitizer in a controlled way which is in ROS-sensitive manner and CD44 receptor mediated.   

Given the use of photosensitizers in cancer treatment, work on synthesis of such compounds is interesting. However, manuscript in general is not well written. Sentence phrasing is confusing at many places. Some of the experiments/readout and not strongly convincing. It would need a thorough revision.

I have following concerns-

  1. Title should be revised. Authors seem to weave many points in a line, and it is creating chaos as reader.

  1. Abstract needs thorough revision. Background, gap, and the aim of research in not clear from the abstract.

  1. Line 21. Please rephrase the sentence for correctness.

  1. Line 23-24. Please rephrase. it seems a repeat. For instance, "HAthCe6 Nano-photosensitizer can be disintegrated". Disintegration in terms of morphology??? right?? This has been already mentioned in previous sentence.

  1. Line 25-26. “Nanophotosensitizers were treated to HeLa human cervical cancer cells" Please revise the sentence.

  1. Line 30-31. Please revise.

  1. Figure 2. Line 152. where is legend for (b)? Do these nano-photosensitizer have aggregation tendency?

  1. Line 167. “then started to disintegration at 5mM H2O2”. Disintegrate as in?? Would disintegration lead to increased size?

  1. Figure 3. Did authors examine the particle size distribution at different concentrations of HAthCe6? Graph panels A(a), (b) and (c) indicate that with increasing H2O2 concentration, particles size distribution varies (particularly increases). Any explanation for this change? It seems to have two different size particles. Could it be due to the agglomeration? However, this not clear in in TEM images?? It would be helpful if author indicate the particles using any arrow in the image. Panel B(a), (b) and (c) look very different and it is not clear what is particle and what not.

  1. Line 175. which particle size? please mention.

  1. Line 187. It appears to be concentration dependent. How would it be controlled? Do authors imply the use of oxidizing agent to increase theCe6 release?

  1. Figure 5(a) How did this fluorescence intensity was determined? Please refer to the comment in method section.

  1. Figure 5(b). These images are not enough to convince the uptake at the low magnification.

  1. Line 214-215. Please revise the sentence.

  1. Line 219. What did authors imply by small intrinsic cytotoxicity?

  1. Line 220-221. “Furthermore, nano-220 photosensitizers have no acute toxicity against normal cells and cancer cells.” Any further evidence for this conclusion?

  1. Figure 6. It is confusing to have Ce6 on the x-axis. Please revise the axis label.

  1. Line 228-232. NPT is sensitive to ROS generation as concluded in Line 172 above. Here experiments are demonstrating that NPT itself has ROS generation potential. Did authors examined ROS generation in any non-cancer cells line upon treating with NTP at different concentrations?

  1. Figure 7B. What is control (as mentioned in Y-axis) here?

  1. Line 439-446. To convince the uptake of NPS, florescence should also be measured without lysis. What is the purpose of dissolving the cells?

  1. In general figure legends should be properly informative about the data/panel/graph presents. Please consider revising throughout the manuscript.

Author Response

Response to Reviewer 1’s comment

In the manuscript IJMS-1591422 by Yoon et al, authors have synthesized HAthCe6, a chlorin e6 (Ce6)-conjugated with hyaluronic acid (HA) and demonstrated its efficacy for reactive oxygen species (ROS)-sensitive and CD44 receptor-targeted delivery of photosensitizers and for photodynamic therapy (PDT) of HeLa cervical carcinoma cells. In this study, authors investigated physicochemical and biological properties of HAthCe6 nano-photosensitizers both in in vitro and in vivo settings. Authors found that the uptake ratio and efficacy of photodynamic therapy of conjugated Ce6 was significantly higher than the free Ce6. Authors found HAthCe6 nano-photosensitizers effective against HeLa cells and proposed the use of conjugated (HAthCe6) nano-photosensitizer in a controlled way which is in ROS-sensitive manner and CD44 receptor mediated.   

Given the use of photosensitizers in cancer treatment, work on synthesis of such compounds is interesting. However, manuscript in general is not well written. Sentence phrasing is confusing at many places. Some of the experiments/readout and not strongly convincing. It would need a thorough revision.

I have following concerns-

  1. Title should be revised. Authors seem to weave many points in a line, and it is creating chaos as reader.

 Answer) Thank you for your valuable comment. According to your comment, we changed the title for readers.

CD44 receptor-mediated/reactive oxygen species-sensitive delivery of nanophotosensitizers against cervical cancer cells

  1. Abstract needs thorough revision. Background, gap, and the aim of research in not clear from the abstract.

 Answer) Thank you for your valuable comment. According to your comment, we revised the abstract section as follows:

The aim of this study is to investigate CD44 receptor and reactive oxygen species (ROS)-sensitive delivery of nanophotosensitizers of chlorin e6 (Ce6)-conjugated hyaluronic acid (HA) against HeLa human cervical cancer cells. For synthesis of nanophotosensitizers, thioketal diamine was conjugated with carboxyl group in HA and then amine end group of HA-thioketal amine conjugates was conjugated again with Ce6 (Abbreviated as HAthCe6). HAthCe6 nanophotosensitizers revealed small diameter with sizes less than 200. Their morphology was round shape in the observations of transmission electron microscope (TEM). HAthCe6 nanophotosensitizers responded to oxidative stress-induced size changes when H2O2 was added to the nanophotosensitizer aqueous solution, i.e. their monomodal distribution pattern at 0 mM H2O2 was changed to dual- and/or multi-modal distribution patterns at higher concentration of H2O2. Furthermore, oxidative stress induced by H2O2 addition contributed to the disintegration of HAthCe6 nanophotosensitizers and then accelerated release rate of Ce6 from nanophotosensitizers. When nanophotosensitizers were treated to HeLa cells, Ce6 uptake ratio, ROS generation and PDT efficacy was significantly increased compared to treatment of free Ce6. Since CD44 receptor of cancer cells specifically binds with HA, pretreatment of HA against HeLa cells decreased Ce6 uptake ratio, ROS generation and PDT efficacy of HAthCe6 nanophotosensitizers. These results indicated that intracellular delivery of HAthCe6 nanophotosensitizers can be controlled by CD44 receptor-mediated pathway. Furthermore, these phenomena induced CD44 receptor-controllable ROS generation and PDT efficacy by HAthCe6 nanophotosensitizers. At in vivo tumor imaging using HeLa cells, nanophotosensitizer administration showed that fluorescence intensity in tumor tissues was relatively higher than those of other organs. When free HA was pretreated, fluorescence intensity of tumor tissue was relatively lower than lower than those of other organs, indicating that HAthCe6 nanophotosensitizers have CD44 receptor sensitivity and then they can be delivered by receptor-specific manner. We suggest that HAthCe6 nanophotosensitizers are promising candidate for PDT of cervical cancer.

  1. Line 21. Please rephrase the sentence for correctness.

 Answer) Thank you for your valuable comment. According to your comment, we revised this sentences.

HAthCe6 nanophotosensitizers responded to oxidative stress-induced changes in size distribution when H2O2 was added to the nanophotosensitizer aqueous solution, i.e. their monomodal distribution pattern at 0 mM H2O2 was changed to dual- and/or multi-modal distribution patterns at higher concentration of H2O2.

  1. Line 23-24. Please rephrase. it seems a repeat. For instance, "HAthCe6 Nano-photosensitizer can be disintegrated". Disintegration in terms of morphology??? right?? This has been already mentioned in previous sentence.

 Answer) Thank you for your valuable comment. According to your comment, we revised this sentences.

Furthermore, oxidative stress induced by H2O2 addition contributed to the disintegration of HAthCe6 nanophotosensitizers in morphology and then this phenomenon accelerated release rate of Ce6 from nanophotosensitizers.

  1. Line 25-26. “Nanophotosensitizers were treated to HeLa human cervical cancer cells" Please revise the sentence.

 Answer) Thank you for your valuable comment. According to your comment, we revised this sentences.

In cell culture study using HeLa cells, nanophotosensitizers increased Ce6 uptake ratio, ROS generation and PDT efficacy compared to free Ce6.

  1. Line 30-31. Please revise.

 Answer) Thank you for your valuable comment. According to your comment, we revised this sentences.

Since HA specifically bonds with CD44 receptor of cancer cells, pretreatment of free HA against HeLa cells decreased Ce6 uptake ratio, ROS generation and PDT efficacy of HAthCe6 nanophotosensitizers.

  1. Figure 2. Line 152. where is legend for (b)? Do these nano-photosensitizer have aggregation tendency?

 Answer) Thank you for your valuable comment. According to your comment, we revised the Figure legend.

Figure 2. HAthCe6 nanophotosensitizers. (a) Particle size distribution and (b) TEM image. Average particle size was 146.1±35.3 nm as described in Table 1. Poly-dispersity index (PDI) was 0.036. TEM images were negative staining of nanophotosensitizers with phosphotungstic acid.

  1. Line 167. “then started to disintegration at 5mM H2O2”. Disintegrate as in?? Would disintegration lead to increased size?

 Answer) Thank you for your valuable comment. Disintegration of HAthCe6 nanophotosnesitizers must be occurred by oxidative stress (induced by hydrogen peroxide) since thioketal group is known to be degraded by oxidative stress. Because Ce6 was linked to the backbone of HA through thioketal linkage, morphology of nanophotosensitizers might became disintegrated. Especially, Ce6 is a hydrophobic segment in the conjugates while HA backbone act as a hydrophilic domain. Then, amphiphilic conjugates such as HAthCe6 can form self-assemblies in the aqueous solution. If hydrophobic segment (Ce6) was liberated from HAthCe6 nanophotosnesitizers, driving force of self-assembling of nanophotosensitizers must be losted in the aqueous solution. Then, nanophotosensitizers must be swelled in the low concentration of hydrogen peroxide and then size must be increased. And, at higher concentration of H2O2, nanophosensitizers must be dis-assembled in the aqueous solution. Anyway, we discussed more about this in the Results and discussion section.

As shown in Figure 3 (a), (b) and (c), monomodal distribution pattern in size distribution at 0 or 1.0 mM H2O2 was changed to dual or multi-modal distribution pattern at higher H2O2 concentration. At low H2O2 concentrations (1.0 mM), size distribution of nanophotosensitizers became broader compared to untreated samples as shown in Figure 2(a) and (b). H2O2 concentration higher than 5 mM resulted in dual or multi-modal distribution patterns, i.e. size distribution was dual-distribution pattern at 5.0 mM H2O2 and, at 10 mM H2O2, measurement was practically fialed (Figure 3 (b) and (c)). These results might be due to that thioketal linkage between HA and Ce6 must be broken at low H2O2 concentration and then Ce6 was separated from the HAthCe6 conjugates. These must be induced increase of particle size distribution. Furthermore, they must be disintegrated at higher low H2O2 concentration by liberation of Ce6 from nanophotosensitizers. Morphological observation supported these results as shown in Figure 3(d), (e) and (f). Morphologies of nanophotosensitizers were changed compared to untreated samples in Figure 2(b), i.e. some of the nanophotosensitizers were started to be swelled and/or disintegrated at low H2O2 concentrations (1.0 mM) even though most of them still maintain their spherical morphology (Figure 3(d)). Furthermore, nanophotosensitizers were largely swelled or disintegrated at 5 mM H2O2 and then, at 10 mM H2O2, most of them was disintegrated as shown in Figure 3(f), indicating that HAthCe6 nanophotosensitizers have ROS-sensitive disintegration properties. Table 2 abbreviated the particle size distribution properties presented at Figure 3(a), (b) and (c).

  1. Figure 3. Did authors examine the particle size distribution at different concentrations of HAthCe6? Graph panels A(a), (b) and (c) indicate that with increasing H2O2 concentration, particles size distribution varies (particularly increases). Any explanation for this change? It seems to have two different size particles. Could it be due to the agglomeration? However, this not clear in in TEM images?? It would be helpful if author indicate the particles using any arrow in the image. Panel B(a), (b) and (c) look very different and it is not clear what is particle and what not.

 Answer) Thank you for your valuable comment. Practically, changes in particle size distribution and morphologies of nanophotosensitizers in Figure 3 is to explain the reason of why Ce6 release rate was increased by addition of H2O2 as shown in 4. As you indicated, some changes in particle size distribution must be observed in different nanohotosensitizer concentration and dual or multi modal distribution can be observed at higher or lower concentration of nanophotosensitizers in water. Also, as you indicated, agglomeration of nanoparticles can be occurred at higher concentration. Practically, we used same sample (and same concentration of nanophotosensitizer solution) to measure particle size at 0, 5 and 10 mM H2O2. Anyway, we discussed about this in the discussion section. Thanks for your valuable comment.

Oxidative stress may degrade thioketal linker of HAthCe6 nanophotosensitizers and then liberated Ce6. Sun et al., also reported that hyperbranched polyphosphate containing thioketal linker can be disintegrated by light-irradiation and then doxorubicin release rate was accelerated [51]. Our results also showed that thioketal linker between HA and Ce6 might be disintegrated in the low concentration of H2O2 and then particle size distribution was increased as shown in Figure 3A(a). At higher H2O2 concentration, Ce6 might be separated from HA backbone of HAthCe6 conjugates and then liberated from nanophotosensitizers as shown in Figure 3 and 4. Chen et al., also reported that oxidative stress resulted in changes of size distribution of thioketal nanoparticles, i.e. size distribution of nanoparticles was changed from monomodal distribution pattern to multimodal distribution pattern by addition of H2O2 [52]. Furthermore, they also showed that H2O2 treatment induces degradation of nanoparticles and then drug release rate was significantly increased.

Chen, D.; Zhang, G.; Li, R.; Guan, M.; Wang, X.; Zou, T.; Zhang, Y.; Wang, C.; Shu, C.; Hong, H.; Wan, L.J. Biodegradable, hydrogen peroxide, and glutathione dual responsive nanoparticles for potential programmable paclitaxel release. J. Am. Chem. Soc. 2018, 140, 7373-7376.

  1. Line 175. which particle size? please mention.

 Answer) Thank you for your valuable comment. According to you comment, we mentioned in the Figure legend as follows:

Figure 3. The effect of H2O2 concentration on the changes in particle size distribution ((a), (b) and (c)) and TEM images ((d), (e) and (f)) of HAthCe6 nanophotosensitizers. (a) and (d), 1.0 mM H2O2; (b) and (e), 5.0 mM H2O2; (c) and (f), 10 mM H2O2.

  1. Line 187. It appears to be concentration dependent. How would it be controlled? Do authors imply the use of oxidizing agent to increase theCe6 release?

 Answer) Thank you for your valuable comment. Oxidizing agent such as H2O2 may contribute to the degradation of thioketal linkage between HA backbone and Ce6. And then, separated Ce6 in the nanophotosensitizers must be easily released to the outside of particles. Practically, in the release experiment, H2O2 was directly added to the drug release media (such as PBS). If nanophotosensitizers are in the biological system of human body, increased oxidative stress in the tumor tissues may accelerate degradation of thioketal linkages of HAthCe6 nanophotosensitizers and, furthermore, light irradiation may increase oxidative stress in the tumor (More exactly, in the field of light irradiation) and then elevated oxidative stress also accelerate disintegration of nanophotosensitizers and Ce6 release. Anyway, we discussed more in the discussion section.

Especially, the fact that ROS level is normally elevated in the tumor tissue can be used as a targeting issue and then this status induces tumor-specific degradation of nanoparticles to liberate anticancer agents [53,54].

  1. Kobayashi, H.; Imanaka, S.; Shigetomi, H. Revisiting therapeutic strategies for ovarian cancer by focusing on redox homeostasis. Lett. 2022, 23, 80.
  2. Liao, J.X.; Huang, Q.F.; Li, Y.H.; Zhang, D.W.; Wang, G.H. Chitosan derivatives functionalized dual ROS-responsive nanocarriers to enhance synergistic oxidation-chemotherapy. Carbohydr Polym. 2022, 282, 119087.

  1. Figure 5(a) How did this fluorescence intensity was determined? Please refer to the comment in method section.

 Answer) Thank you for your valuable comment. According to your comment, we indicated this in the method section.

Figure 5. Ce6 uptake ratio of HeLa cells. (a) HA pretreatment effect on the Ce6 uptake ratio. (b) Fluorescence images of HeLa cells. Free HA (1.0 or 5.0 mg/ml) was pretreated to HeLa cells 30 min before treatment of nanophotosensitizers. Cells were treated with Ce6 or nanophotosensitizers for 2 h and then cells were lysed to measure intracellular Ce6 level. Fluorescence intensity was measured at 407 nm excitation wavelength and 664 nm of emission wavelength using Infinite M200 pro microplate reader. Magnification of images: 100 ×.

In the method section

4.10. Intracellular uptake of free Ce6 or nanophotosensitizers

2×104 HeLa cells seeded in 96 well plates were cultured overnight at 37oC in 5% CO2. Ce6 or HAthCe6 nanophotosensitizers were treated to cells for 2 h and then cells were washed with PBS. These were lysed to measure intracellular Ce6 level. GenDEPOT lysis buffer (100 µl) (Barker, TX, USA) was used for lysis of cells. Infinite M200 pro microplate reader, Tecan Trading AG (Männedorf, Switzerland) was used to measure Ce6 (λex = 407; λem = 664 nm).

To study the effect of CD44 receptor blocking, free HA (1.0 or 5.0 mg/ml) was pretreated to HeLa cells 30 min before nanophotosensitizer treatment. After that, cells were treated with Ce6 or nanophotosensitizers for 2 h and then cells were lysed as decribed above. Fluorescence intensity was measured at 407 nm excitation wavelength and 664 nm of emission wavelength using Infinite M200 pro microplate reader.

  1. Figure 5(b). These images are not enough to convince the uptake at the low magnification.

 Answer) Thank you for your valuable comment. At this moment, we modified the images to confirm cell images better. These images are supported to the results of Figure 5(a). We added higher magnification images in the supplementary files.

Figure S2. HeLa cells observed with confocal scanning microscope. The cells were pretreated with folic acid as similar to Figure 5 and, after that, the nanophotosensitizers were treated. Magnification: 400 ×.

  1. Line 214-215. Please revise the sentence.

 Answer) Thank you for your valuable comment.

As well as free Ce6, nanophotosensitizers did not significantly affect to the viability of HeLa cells and then 80 % of cells were viable until 5 µg/ml (Ce6 equivalent).

  1. Line 219. What did authors imply by small intrinsic cytotoxicity?

 Answer) Thank you for your valuable comment. In this study, we hypothesized that 80 % of cell viability can be considered as a low cytotoxicity.

These results indicated that nanophotosensitizers have reduced cytotoxicity both RAW264.7 and HeLa cells as well as similar to free Ce6. Furthermore, nanophotosensitizers have no acute cytotoxicity against normal cells and cancer cells.

  1. Line 220-221. “Furthermore, nano-220 photosensitizers have no acute toxicity against normal cells and cancer cells.” Any further evidence for this conclusion?

 Answer) Thank you for your valuable comment. At this moment, we have dark toxicity results. Practically, nanophotosensitizers are concentrated in the tumor rather than free Ce6 and then revealed lower cytotoxicity against normal cells or tumor cells (in the absence of light irradiation). Thanks.

  1. Figure 6. It is confusing to have Ce6 on the x-axis. Please revise the axis label.

 Answer) Thank you for your valuable comment.

Figure 6. Dark toxicity of free Ce6 and nanophotosensitizers. (a) RAW264.7 cell; (b) HeLa cells.

  1. Line 228-232. NPT is sensitive to ROS generation as concluded in Line 172 above. Here experiments are demonstrating that NPT itself has ROS generation potential. Did authors examined ROS generation in any non-cancer cells line upon treating with NTP at different concentrations ?

 Answer) Thank you for your valuable comment. In this study, we conformed that NPT has also capacity to generate ROS in the light irradiation (if light irradiation was absence, NPT did not produce ROS as similar to Ce6). Practically, (even though we did not show the results) NPT under light irradiation produces ROS in non-cancer cells (Such as RAW264.7 cells and CCD986Sk cells) and revealed phototoxicity. This is the reason why PDT application is limited to the local region (having tumor). At this moment, we are going to do separate experiment for intrinsic toxicity, ROS formation and phototoxicity against normal cells in vitro and animal (mouse) in vivo to evaluate safety of HAthCe6 nanophotosensitizers. Thanks again for your valuable advise.

  1. Figure 7B. What is control (as mentioned in Y-axis) here?

 Answer) Thank you for your valuable comment. For control treatment, serum free media were used instead of Ce6 or nanophotosensitizer solution. We indicated it in the figure legend.

Figure 7. The effect of HAthCe6 nanophotosensitizers against HeLa cells. (a) ROS generation; (b) PDT efficacy. Free Ce6 or nanophotosensitizers in serum-free media was treated to 2×104 cells/well in 96 well plates and then irradiated at light dose of 2 J/cm2. DCFH-DA assay was used to evaluate intracellular ROS level. For control treatment, serum-free media were used.

  1. Line 439-446. To convince the uptake of NPS, florescence should also be measured without lysis. What is the purpose of dissolving the cells?

 Answer) Thank you for your valuable comment. To measure intracellular Ce6 concentration, cells should be dissolved in the lysis buffer because Ce6 concentration in live cells must be distorted compared to dissolving the cells. In fact, control cells (without treatment of free Ce6 or NPT) was used to zero test since cells have their intrinsic fluorescence intensity (even though auto fluorescence of cells are very low).

  1. In general figure legends should be properly informative about the data/panel/graph presents. Please consider revising throughout the manuscript.

 Answer) Thank you for your valuable comment. According to your comment, we revised the Figure legend to be informative the data. Thanks a lot.

Figure 1. Synthesis scheme (Left) and 1H NMR spectra (Right) of HAthCe6 conjugates. Dimethyl sulfoxide (DMSO)-d form was used to dissolve Ce6 or thioketal diamine and dimethyl sulfoxide (DMSO)-d form/D2O (4/1, v/v) mixtures were used to dissolve HA or HAthCe6 conjugates.

Figure 2. HAthCe6 nanophotosensitizers. (a) Particle size distribution and (b) TEM image. Average particle size was 146.1±35.3 nm as described in Table 1. Poly-dispersity index (PDI) was 0.036. For TEM images, nanophotosensitizers were negatively stained with phosphotungstic acid.

Figure 3. The effect of H2O2 concentration on the changes in particle size distribution ((a), (b) and (c)) and TEM images ((d), (e) and (f)) of HAthCe6 nanophotosensitizers. (a) and (d), 1.0 mM H2O2; (b) and (e), 5.0 mM H2O2; (c) and (f), 10 mM H2O2.

Figure 4. The effect of H2O2 addition on the aqueous nanophotosensitizer solution. (a) Ce6 release from nanophotosensitizers. (b) Changes of fluorescence spectra of nanophotosensitizers in aqueous solution. For measurement of fluorescence intensity, the concentration of nanophotosensitizer solution was adjusted to 1.0 mg/ml in PBS. H2O2 was added to release media for Ce6 release study.  For fluorescence imaging, nanophotosensitizer solution was incubated for 2h at 37oC in the absence or presence of different concentration of H2O2.

Figure 5. Ce6 uptake ratio of HeLa cells. (a) HA pretreatment effect on the Ce6 uptake ratio. (b) Fluorescence images of HeLa cells. Free HA (1.0 or 5.0 mg/ml) was pretreated to HeLa cells 30 min before treatment of nanophotosensitizers. Cells were treated with Ce6 or nanophotosensitizers for 2 h and then cells were lysed to measure intracellular Ce6 level. Fluorescence intensity was measured at 407 nm excitation wavelength and 664 nm of emission wavelength using Infinite M200 pro microplate reader. Magnification of images: 100 ×.

Figure 6. Dark toxicity of free Ce6 and nanophotosensitizers. (a) RAW264.7 cell; (b) HeLa cells. Cells were treated with free Ce6 or HAthCe6 nanophotosensitizers for 2 h without light irradiation. For control treatment, serum-free media were used for comparison.

Figure 7. The effect of HAthCe6 nanophotosensitizers against HeLa cells. (a) ROS generation; (b) PDT efficacy. Free Ce6 or nanophotosensitizers in serum-free media was treated to 2×104 cells/well in 96 well plates and then irradiated at light dose of 2 J/cm2. DCFH-DA assay was used to evaluate intracellular ROS level. For control treatment, serum-free media were used.

Figure 8. in vivo tumor imaging of HeLa tumor-bearing mice. 30 min before administration of nanophotosensitizers, free HA (10 mg/kg) was i.v. administered via tail vein of mice. Then, nanophotosensitizers (10 mg/kg) were i.v. administered. 24 h later, mice were sacrificed to observe fluorescence imaging of organs using fluorescence imaging device (Maestro II®).

Reviewer 2 Report

The manuscript presented by Jieun Yoon et al., describes the synthesis and application of a new nanophotosensitizer for targeted PDT of cervical cancer cells that employs HA and Ce6 as targeting and photosensitizing units, respectively, conjugated by H2O2-sensitive linkage. Despite the sound idea and the competent experimental work, the presentation of the results has serious flaws. Starting with the confusing long title and rather unfocused abstract, the presented work is not able to attract readers attention.

 The main problem is the rather poor and unclear description of the results. The quality of figures (esp. Fig. 1 and Fig.4) is not sufficient and the information they intend to present is not clear - no proper numbering of the given spectra, or nor spectra shown although stated in caption of Fig.4b. 

The purpose of Table 1 is not clear. The estimation of the w/w % of Ce6 in the nano-conjugate by simple measurements of the fluorescence spectra of free and released Ce6 (section 4.4) needs more details with exact concentration after the stated dilutions (more than 10 times is not informative). A series of measurements and/or calibration curve might be needed for better estimation. The details about the Ce6 content are important because in the following comparisons in the in vitro tests the effects are estimated at certain Ce6 concentration (in mg/ml) and it must be clearly stated how this concentration is calculated for the nanocomposite amount used for the treatment (which could be given in the caption of the figures). 

What is the reason to compare the effects on RAW264.7 cells and HeLa cells remains unclear.

The discussion has very little connection with the described results and must be rewritten. The same holds for most of the parts in the manuscript. The abstract should highlight the necessity of the work and the importance of the obtained result, rather than listing the performed experiments.

My overall conclusion is that the presented manuscript cannot be accepted in its present form - it has to be rewritten and additional control experiments for determining the concentration/content of Ce6 in the NPT should be provided. Resubmission could be an option.

Author Response

The manuscript presented by Jieun Yoon et al., describes the synthesis and application of a new nanophotosensitizer for targeted PDT of cervical cancer cells that employs HA and Ce6 as targeting and photosensitizing units, respectively, conjugated by H2O2-sensitive linkage. Despite the sound idea and the competent experimental work, the presentation of the results has serious flaws. Starting with the confusing long title and rather unfocused abstract, the presented work is not able to attract readers attention.

Answer) Thank you for your valuable comment. We revised the manuscript according to your comment. Please consider again. Thanks.

 The main problem is the rather poor and unclear description of the results. The quality of figures (esp. Fig. 1 and Fig.4) is not sufficient and the information they intend to present is not clear - no proper numbering of the given spectra, or nor spectra shown although stated in caption of Fig.4b. 

Answer) Thank you for your valuable comment. According to your comment, we revised the Figures and some of errors were corrected.

Figure 1. Synthesis scheme (Left) and 1H NMR spectra (Right) of HAthCe6 conjugates. Dimethyl sulfoxide (DMSO)-d form was used to dissolve Ce6 or thioketal diamine and dimethyl sulfoxide (DMSO)-d form/D2O (4/1, v/v) mixtures were used to dissolve HA or HAthCe6 conjugates.

Figure 4. The effect of H2O2 addition on the aqueous nanophotosensitizer solution. (a) Ce6 release from nanophotosensitizers. (b) Changes of fluorescence spectra of nanophotosensitizers in aqueous solution. For measurement of fluorescence intensity, the concentration of nanophotosensitizer solution was adjusted to 1.0 mg/ml in PBS. H2O2 was added to release media for Ce6 release study.  For fluorescence imaging, nanophotosensitizer solution was incubated for 2h at 37oC in the absence or presence of different concentration of H2O2.

The purpose of Table 1 is not clear. The estimation of the w/w % of Ce6 in the nano-conjugate by simple measurements of the fluorescence spectra of free and released Ce6 (section 4.4) needs more details with exact concentration after the stated dilutions (more than 10 times is not informative). A series of measurements and/or calibration curve might be needed for better estimation. The details about the Ce6 content are important because in the following comparisons in the in vitro tests the effects are estimated at certain Ce6 concentration (in mg/ml) and it must be clearly stated how this concentration is calculated for the nanocomposite amount used for the treatment (which could be given in the caption of the figures). 

Answer) Thank you for your valuable comment. According to your comment, we calculated Ce6 from the calibration curve from UV-VIS spectrophotometer and fluorescence spectrophotometer.  

Table 1. Characterization of HAthCe6 conjugates

Drug contents (%, w/w)

Particle size (nm)

Theoreticala

Experimentalb

HAthCe6 conjugates

10.3

9.3

146.1±35.3

a Theoretical content was calculated from Feeding weight of Ce6.

b Experimental content was measured as depicted in Materials and methods section and supplementary materials. Ce6 cocentration in the release media was calculated from the calibration curve of Ce6 (Figure S1).

Figure S1. Calibration curve of Ce6. (a) DMSO/water, 9/1 (v/v); (b) Water. For calibration curve in DMSO, Ce6 (1mg/ml DMSO) was diluted with DMSO/water mixed solvent (9/1, v/v). For calibration curve, absorbance of this solution was measured with Genesys 10s UV-VIS spectrophotometer at 664 nm. Calibration curve was measured at the range of 0.1 ~ 7 µg/ml of Ce6 concentration. For calibration curve in PBS, Ce6 (1mg/ml DMSO) was diluted with PBS more than 30 times. Calibration curve was measured at the range of 0.3 ~ 10 µg/ml of Ce6 concentration.

Calibration curve using fluorescence spectrophotometer (data not shown).

What is the reason to compare the effects on RAW264.7 cells and HeLa cells remains unclear.

Answer) Thank you for your valuable comment. In fact, we wanted to evaluate the intrinsic/dark toxicity of HAthCe6 nanophotosensitizers both normal cells and cancer cells. This means that, in the absence of light irradiation, nanophotosensitizers have no cytotoxicity against normal cells (because light irradiation will be adapted to the specific site of action) as well as free Ce6.

The discussion has very little connection with the described results and must be rewritten. The same holds for most of the parts in the manuscript. The abstract should highlight the necessity of the work and the importance of the obtained result, rather than listing the performed experiments.

Answer) Thank you for your valuable comment. According to your comment, we revised the manuscript. Abstract, introduction and discussion was rewritten/discussed to explain the data in the discussion section.

Abstract: Stimuli-sensitive nanomedicine-based photosensitizer delivery has an opportunity to target tumor tissues since oxidative stress and expression of molecular proteins such as CD44 receptors is elevated in the tumor microenvironment. The aim of this study is to investigate CD44 receptor and reactive oxygen species (ROS)-sensitive delivery of nanophotosensitizers of chlorin e6 (Ce6)-conjugated hyaluronic acid (HA) against HeLa human cervical cancer cells. For synthesis of nanophotosensitizers, thioketal diamine was conjugated with carboxyl group in HA and then amine end group of HA-thioketal amine conjugates was conjugated again with Ce6 (Abbreviated as HAthCe6). HAthCe6 nanophotosensitizers revealed small diameter with sizes less than 200. Their morphology was round shape in the observations of transmission electron microscope (TEM). HAthCe6 nanophotosensitizers responded to oxidative stress-induced changes in size distribution when H2O2 was added to the nanophotosensitizer aqueous solution, i.e. their monomodal distribution pattern at 0 mM H2O2 was changed to dual- and/or multi-modal distribution patterns at higher concentration of H2O2. Furthermore, oxidative stress induced by H2O2 addition contributed to the disintegration of HAthCe6 nanophotosensitizers in morphology and then this phenomenon accelerated release rate of Ce6 from nanophotosensitizers. In cell culture study using HeLa cells, nanophotosensitizers increased Ce6 uptake ratio, ROS generation and PDT efficacy compared to free Ce6. Since HA specifically bonds with CD44 receptor of cancer cells, pretreatment of free HA against HeLa cells decreased Ce6 uptake ratio, ROS generation and PDT efficacy of HAthCe6 nanophotosensitizers. These results indicated that intracellular delivery of HAthCe6 nanophotosensitizers can be controlled by CD44 receptor-mediated pathway. Furthermore, these phenomena induced CD44 receptor-controllable ROS generation and PDT efficacy by HAthCe6 nanophotosensitizers. At in vivo tumor imaging using HeLa cells, nanophotosensitizer administration showed that fluorescence intensity in tumor tissues was relatively higher than those of other organs. When free HA was pretreated, fluorescence intensity of tumor tissue was relatively lower than lower than those of other organs, indicating that HAthCe6 nanophotosensitizers have CD44 receptor sensitivity and then they can be delivered by receptor-specific manner. We suggest that HAthCe6 nanophotosensitizers are promising candidate for PDT of cervical cancer.

In discussion section

Oxidative stress may degrade thioketal linker of HAthCe6 nanophotosensitizers and then liberated Ce6. Sun et al., also reported that hyperbranched polyphosphate containing thioketal linker can be disintegrated by light-irradiation and then doxorubicin release rate was accelerated [51]. Our results also showed that thioketal linker between HA and Ce6 might be disintegrated in the low concentration of H2O2 and then particle size distribution was increased as shown in Figure 3A(a). At higher H2O2 concentration, Ce6 might be separated from HA backbone of HAthCe6 conjugates and then liberated from nanophotosensitizers as shown in Figure 3 and 4. Chen et al., also reported that oxidative stress resulted in changes of size distribution of thioketal nanoparticles, i.e. size distribution of nanoparticles was changed from monomodal distribution pattern to multimodal distribution pattern by addition of H2O2 [52]. Furthermore, they also showed that H2O2 treatment induces degradation of nanoparticles and then drug release rate was significantly increased. Especially, the fact that ROS level is normally elevated in the tumor tissue can be used as a targeting issue and then this status induces tumor-specific degradation of nanoparticles to liberate anticancer agents [53,54].

Our results also showed that intracellular delivery of HAthCe6 nanophotosensitizers was inhibited by blocking of CD44 receptor of cancer cells as shown in Figure 5. These phenomena induced CD44 receptor-dependent ROS generation and phototoxicity of HAthCe6 nanophotosensitizers in HeLa cells as shown in Figure 7. These results indicated that HAthCe6 nanophotosensitizers has responsiveness against CD44 receptor of HeLa cells and then therapeutic potential can be controlled by receptor expression. Furthermore, CD44 receptor-mediated delivery of HAthCe6 nanophotosensitizers was also inhibited by CD44 receptor blocking of HeLa tumor at in vivo animal model as shown in Figure 8. Our results also showed that HAthCe6 nanophotosensitizers can be delivered through ROS-sensitive and CD44 receptor-mediated manner. Especially, nanophotosensitizers were efficiently concentrated in the tumor tissue, i.e fluorescence intensity by i.v. administration of nanophotosensitizers was strongest in tumor tissue as shown in Figure 8. This peculiarity of HAthCe6 nanophotosensitizers may alleviate light sensitivity of normal tissues and PDT efficacy against tumor. HA-decorated nanoparticles can be delivered to cancer cells with CD44 receptor-specific manner and delivering capacity was accelerated against CD44-receptor over-expressing cells [59]. Kim et al., also HA-decorated nanophotosensitizers selectively targeted CD44-receptor positive cells and then killed the cancer cells with CD-responsive manner [60]. They argued that PDT efficacy of HA-decorated nanophotosensitizers was controlled by CD44 receptor expression while CD44 receptor negative cells were not affected by blocking of CD44 receptor. Delivery capacity and PDT efficacy of our HAthCe6 nanophotosensitizers was also easily controlled by CD44-receptor positive cells. These results indicated that HAthCe6 nanophotosensitizers can be used for specific PDT of cervical cancer with minimization of side-effects against normal ells since cancer cells are overexpressed CD44 receptor [61].

  1. Wang, R.; Yang, H.; Khan, A.R.; Yang, X.; Xu, J.; Ji, J.; Zhai, G. Redox-responsive hyaluronic acid-based nanoparticles for targeted photodynamic therapy/chemotherapy against breast cancer. Colloid Interface Sci. 2021, 598, 213-228.
  2. Kim, D.M.; Shim, Y.H.; Kwon, H.; Kim, J.P.; Park, J.I.; Kim, D.H.; Kim, D.H.; Kim, J.H.; Jeong, Y.I. CD44 receptor-specific and redox-sensitive nanophotosensitizers of hyaluronic acid-chlorin e6 tetramer having diselenide linkages for photodynamic treatment of cancer cells. Pharm. Sci. 2019, 108, 3713-3722. 
  3. Chen, C.; Zhao, S.; Karnad, A.; Freeman, J.W. The biology and role of CD44 in cancer progression: therapeutic implications. Hematol. Oncol. 2018, 11, 64.

My overall conclusion is that the presented manuscript cannot be accepted in its present form - it has to be rewritten and additional control experiments for determining the concentration/content of Ce6 in the NPT should be provided. Resubmission could be an option.

Answer) Thank you for your valuable comment. We revised the manuscript according to your comment. Please re-consider our manuscript for publication in your journal. I appreciated your consideration.

Round 2

Reviewer 1 Report

The revised looks a bit improved.

Authors have used “significant” in most of their result sections. However, there is no information about any statistical test used and p values throughout the manuscript.

Given that present study has a good fraction of quantitative data, it is important to provide the statistical information. Please add the statistical analysis section under the “Materials and Methods”. Provide p value information including the type of test/post hoc used, in graphs and figure legends.

Author Response

Authors have used “significant” in most of their result sections. However, there is no information about any statistical test used and p values throughout the manuscript.

Given that present study has a good fraction of quantitative data, it is important to provide the statistical information. Please add the statistical analysis section under the “Materials and Methods”. Provide p value information including the type of test/post hoc used, in graphs and figure legends.

Answer) Thanks for your comment. According to your comment, we added statistical analysis for results using Sigmaplot program (version: 11.0). And the analysis section was added to Materials and methods. Thanks again.

4.13. Statistical analysis

The statistical analysis of the results was evaluated with Student’s t test using SigmaPlot® program, version: 11.0, Systat Software, Inc. Co. (San Jose, CA, USA). Then, p < 0.01 was evaluated as the minimal level of significance.

Figure 4. The effect of H2O2 addition on the aqueous nanophotosensitizer solution. (a) Ce6 release from nanophotosensitizers. (b) Changes of fluorescence spectra of nanophotosensitizers in aqueous solution. For measurement of fluorescence intensity, the concentration of nanophotosensitizer solution was adjusted to 1.0 mg/ml in PBS. H2O2 was added to release media for Ce6 release study. For fluorescence imaging, nanophotosensitizer solution was incubated for 2h at 37oC in the absence or presence of different concentration of H2O2. Ce6 concentration in the release media was calculated from the calibration curve of Ce6 (Figure S1). *,**,***: p < 0.001.

Figure 5. Ce6 uptake ratio of HeLa cells. (a) HA pretreatment effect on the Ce6 uptake ratio. (b) Fluorescence images of HeLa cells. Free HA (1.0 or 5.0 mg/ml) was pretreated to HeLa cells 30 min before treatment of nanophotosensitizers. Cells were treated with Ce6 or nanophotosensitizers for 2 h and then cells were lysed to measure intracellular Ce6 level. Fluorescence intensity was measured at 407 nm excitation wavelength and 664 nm of emission wavelength using Infinite M200 pro microplate reader. Magnification of images: 100 ×. *,**: p < 0.001; ***: p < 0.01

Figure 6. Dark toxicity of free Ce6 and nanophotosensitizers. (a) RAW264.7 cell; (b) HeLa cells. Cells were treated with free Ce6 or HAthCe6 nanophotosensitizers for 2 h without light irradiation. For control treatment, serum-free media were used for comparison. *,**: p < 0.01

Figure 7. The effect of HAthCe6 nanophotosensitizers against HeLa cells. (a) ROS generation; (b) PDT efficacy. Free Ce6 or nanophotosensitizers in serum-free media was treated to 2×104 cells/well in 96 well plates and then irradiated at light dose of 2 J/cm2. DCFH-DA assay was used to evaluate intracellular ROS level. For control treatment, serum-free media were used. *,**: p < 0.001; ***: p < 0.01. . #: p < 0.01; ##,###: p < 0.001.

Reviewer 2 Report

The revised manuscript follows my recommendations. It seems this is the best out of these results.

Author Response

Response to Reviwer 2's comment

The revised manuscript follows my recommendations. It seems this is the best out of these results.

Answer) Thanks for your kind response. I did my best for your comment. I appreciated to your consideration.

Round 3

Reviewer 1 Report

Considering the importance of repeatability and reproducibility of the data, it is important to perform appropriate and accurate statistical analysis on the acquired data. Please revise the statistical analysis according to the following points.

Two important issues-

  1. Authors are using more than one group (different concentrations and time points). In this case, statistics should be performed using Analysis of Variance (ANOVA, One way or Two way depending on the experiments) rather than student t-test.
  2. Use of * and # is haphazard. Please use standard p value notions. for instance, why "*" and "**"both are ascribed as p<0.001. Number of these asterisks indicate the p value trend. *p<.05, **p<0.01, ***p<0.001 etc. Please arrange the data accordingly. 

I would recommend authors to revise the above mentioned points thoroughly.  

Author Response

Answer to Review’s comment.

  1. Authors are using more than one group (different concentrations and time points). In this case, statistics should be performed using Analysis of Variance (ANOVA, One way or Two way depending on the experiments) rather than student t-test.

Answer) Thanks for your comment. Practically, we learned Prism program to analysis one way ANOVa test and then indicated it in the Materials and methods section and Results section.

4.13. Statistical analysis

The results are expressed as mean ± standard deviation (SD) of three experiments. The statistical analysis of the results was evaluated with Student’s t test using SigmaPlot® program, version: 11.0, Systat Software, Inc. Co. (San Jose, CA, USA) or one-way analysis of variance (ANOVA) followed by the Tukey test using GraphPad Prism 9 (GraphPad Software LLC., CA, USA). Then, p < 0.05 was evaluated as the minimal level of significance.

Figure 4. The effect of H2O2 addition on the aqueous nanophotosensitizer solution. (a) Ce6 release from nanophotosensitizers. (b) Changes of fluorescence spectra of nanophotosensitizers in aqueous solution. For measurement of fluorescence intensity, the concentration of nanophotosensitizer solution was adjusted to 1.0 mg/ml in PBS. H2O2 was added to release media for Ce6 release study. For fluorescence imaging, nanophotosensitizer solution was incubated for 2h at 37oC in the absence or presence of different concentration of H2O2. Ce6 concentration in the release media was calculated from the calibration curve of Ce6 (Figure S1). * : This indicated statistical significance when H2O2 (10 mM) was compared to the H2O2 (0 mM) (ANOVA followed by Turkey’s test, p < 0.05).

Figure 5. Ce6 uptake ratio of HeLa cells. (a) HA pretreatment effect on the Ce6 uptake ratio. (b) Fluorescence images of HeLa cells. Free HA (1.0 or 5.0 mg/ml) was pretreated to HeLa cells 30 min before treatment of nanophotosensitizers. Cells were treated with Ce6 or nanophotosensitizers for 2 h and then cells were lysed to measure intracellular Ce6 level. Fluorescence intensity was measured at 407 nm excitation wavelength and 664 nm of emission wavelength using Infinite M200 pro microplate reader. Magnification of images: 100 ×. ). * : This indicated statistical significance when group of (NPT + free HA, 0 mg/ml) was compared to the group of Ce6 (ANOVA followed by Turkey’s test, p < 0.05).

Figure 7. The effect of HAthCe6 nanophotosensitizers against HeLa cells. Free Ce6 or nanophotosensitizers in serum-free media was treated to 2×104 cells/well in 96 well plates and then irradiated at light dose of 2 J/cm2. (a) ROS generation. DCFH-DA assay was used to evaluate intracellular ROS level. For control treatment, serum-free media were used. (b) PDT efficacy. Cell viability was evaluated by MTT assay. * : This indicated statistical significance when group of (NPT + free HA, 0 mg/ml) was compared to the group of Ce6 (ANOVA followed by Turkey’s test, p < 0.05).

  1. Use of * and # is haphazard. Please use standard p value notions. for instance, why "*" and "**"both are ascribed as p<0.001. Number of these asterisks indicate the p value trend. *p<.05, **p<0.01, ***p<0.001 etc. Please arrange the data accordingly.

Answer) Thanks for your comment. According to your comment, we revised the manuscript and expression of statistical analysis was unified.

Figure 4. The effect of H2O2 addition on the aqueous nanophotosensitizer solution. (a) Ce6 release from nanophotosensitizers. (b) Changes of fluorescence spectra of nanophotosensitizers in aqueous solution. For measurement of fluorescence intensity, the concentration of nanophotosensitizer solution was adjusted to 1.0 mg/ml in PBS. H2O2 was added to release media for Ce6 release study. For fluorescence imaging, nanophotosensitizer solution was incubated for 2h at 37oC in the absence or presence of different concentration of H2O2. Ce6 concentration in the release media was calculated from the calibration curve of Ce6 (Figure S1). * : This indicated statistical significance when H2O2 (10 mM) was compared to the H2O2 (0 mM) (ANOVA followed by Turkey’s test, p < 0.05).

Figure 5. Ce6 uptake ratio of HeLa cells. (a) HA pretreatment effect on the Ce6 uptake ratio. (b) Fluorescence images of HeLa cells. Free HA (1.0 or 5.0 mg/ml) was pretreated to HeLa cells 30 min before treatment of nanophotosensitizers. Cells were treated with Ce6 or nanophotosensitizers for 2 h and then cells were lysed to measure intracellular Ce6 level. Fluorescence intensity was measured at 407 nm excitation wavelength and 664 nm of emission wavelength using Infinite M200 pro microplate reader. Magnification of images: 100 ×. ). * : This indicated statistical significance when group of (NPT + free HA, 0 mg/ml) was compared to the group of Ce6 (ANOVA followed by Turkey’s test, p < 0.05).

Figure 7. The effect of HAthCe6 nanophotosensitizers against HeLa cells. Free Ce6 or nanophotosensitizers in serum-free media was treated to 2×104 cells/well in 96 well plates and then irradiated at light dose of 2 J/cm2. (a) ROS generation. DCFH-DA assay was used to evaluate intracellular ROS level. For control treatment, serum-free media were used. (b) PDT efficacy. Cell viability was evaluated by MTT assay. * : This indicated statistical significance when group of (NPT + free HA, 0 mg/ml) was compared to the group of Ce6 (ANOVA followed by Turkey’s test, p < 0.05).

I would recommend authors to revise the above mentioned points thoroughly.  

Round 4

Reviewer 1 Report

Authors have addressed the concerns.